# A Prospective Evaluation of the Effects of the COVID-19 Pandemic on Youth with Primary Headache Disorders

**DOI:** 10.3390/children10020184

**Published:** 2023-01-19

**Authors:** Mark Connelly, Jennifer Dilts, Madeline Boorigie, Trevor Gerson

**Affiliations:** 1Children’s Mercy Kansas City, Kansas City, MO 64108, USA; 2School of Medicine, University of Missouri-Kansas City, Kansas City, MO 64108, USA

**Keywords:** headache, migraine, stress, coping, children, lifestyle, COVID-19

## Abstract

Alterations in certain academic and social/family routines during the COVID-19 pandemic have been speculated to be either a risk factor or buffer for poor health outcomes for youth with stress-sensitive health conditions such as primary headache disorders. The current study evaluated patterns and moderators of pandemic impacts on youth with primary headache disorders, with an aim of extending our understanding of the relationship between stress, resilience, and outcomes in this population. Children recruited from a headache clinic in the midwestern United States reported on their headaches, schooling, routines, psychological stress, and coping at four timepoints ranging from within a few months of the pandemic onset to a long-term follow-up 2 years later. Changes in headache characteristics over time were analyzed for association with demographics, school status, altered routines, and stress, and coping. At baseline, 41% and 58% of participants reported no change in headache frequency or intensity, respectively, relative to pre-pandemic levels, with the remainder almost equally divided between reporting an improvement or worsening. The results of multilevel growth model analyses indicated that headache intensity remained more elevated over time since the start of the pandemic for respondents whose stress scores were relatively higher (*b* = 0.18, *t* = −2.70, *p* = 0.01), and headache-related disability remained more elevated over time for older respondents (*b* = 0.01, *t* = −2.12, *p* = 0.03). The study results suggest, overall, that the outcomes of primary headache disorders in youth were not systematically altered by the COVID-19 pandemic.

## 1. Introduction

Since the first outbreak of coronavirus disease 2019 (COVID-19) was reported [1], the virus has spread across the world and prompted mitigation measures that have varied in duration and degree by country and region [2]. In addition to potential worries over the health effects of oneself or a family member contracting COVID-19, mitigation measures such as postponement or alterations in schooling and limits on in-person socialization significantly disrupted routines and created additional challenges for young people [3]. Emerging research on the impact of the COVID-19 pandemic on youth in the United States and globally has found trends of increasing stress, anxiety, depression, sleep problems, and suicide risk, with these concerns seemingly amplified in older children, those with pre-existing mental health conditions, and in minority and financially disadvantaged groups [4,5,6,7,8,9,10]. However, negative impacts of the pandemic have not been uniformly observed. For example, some studies have found a subset of children who instead reported higher well-being and life satisfaction early on during the pandemic [11,12,13]. Consistent with coping and stress buffering theories [14,15], children’s coping strategies and their level of perceived family and peer support are speculated to be important moderators of the pandemic’s impact [8,9,13].

Primary headache disorders in children and adolescents, including tension-type and migraine headaches, are health conditions for which symptom frequency and severity are thought to be moderated by stress, lifestyle factors (e.g., sleep, sedentariness, screen time), and mood [16,17]. The extent to which the COVID-19 pandemic has adversely impacted youth with primary headache disorders has, therefore, generated speculation among those who treat these conditions. From one perspective, the disruption to school and family routines, the increase in screen time with virtual or hybrid schooling, and the temporary reductions in access to certain in-person procedural treatments that can be effective in treating some types of primary headache (e.g., pericranial nerve blocks) would be expected to worsen headache frequency and intensity. By contrast, changes that occurred with COVID-19 precautionary measures arguably could be ameliorative for some youth with headaches. For example, the change in schooling to being fully or partly carried out online rather than in-person might reduce school-related difficulties and lessen exposure to environmental headache triggers (fluorescent lights, loud noises) for some children, leading to stable or perhaps even better headache outcomes. Indeed, a cross-sectional study of Italian schoolchildren with primary headache disorders (episodic or chronic migraines and episodic tension-type headaches) conducted early in the pandemic observed a perceived improvement in headaches after the lockdowns for COVID-19, which the researchers attributed to reduction in school-related stress [18]. Similarly, another Italian cross-sectional study of youth with migraines and a US-based study that, in part, included a sample of youth with chronic headaches reported trends of stable or improved headaches during the initial pandemic lockdown periods [19,20].

Investigating the effect of the COVID-19 pandemic on youth with primary headache disorders can be informative for understanding the extent to which lifestyle disruptions have a sustained impact on this population and for understanding variables that moderate this impact, with findings potentially generalizable to other pediatric conditions known to be affected by stress and lifestyle factors. For the current study, we surveyed youth with primary headache disorders at a headache center in the United States during the early and later phases of the pandemic to determine the impact of the pandemic on activities, stress, and headache characteristics. We hypothesized that, on average, headaches would be perceived to be improved at least initially during the pandemic but would worsen for those reporting relatively higher stress and poorer coping efficacy.

## 2. Materials and Methods

### 2.1. Procedure

The study was conducted at Children’s Mercy Kansas City, a freestanding pediatric hospital that serves children from several states in the Midwest region of the United States. The study was approved by the Children’s Mercy Institutional Review Board (IRB #00001335). Participants for this study were recruited from a clinical registry of patients who had been seen in neurology clinics for an evaluation of headaches and who had been diagnosed with a primary headache disorder (e.g., episodic or chronic migraines with or without aura, episodic or chronic tension-type headaches) by a board-certified neurologist or other provider trained in headache medicine (pediatrician or nurse practitioner). In late May 2020 (approximately 3 months after the declaration of a pandemic and prior to the emergence of dominant SARS-CoV-2 variants), an email with information about the study and a link to an online study survey was sent to 2019 families from the registry (“T0”). For families that completed the initial survey, a link to a follow-up survey was sent at 2 months (start of the next school year after pandemic onset, “T1”) and at 4 months (mid-point of the school semester, “T2”) from the date of the initial survey; these timepoints also predated the emergence of dominant SARS-CoV-2 variants. A final longer-term follow-up survey link was sent to respondents at the end of the subsequent school year (24 months from initial survey, “T3”), a time at which the Delta variant (B.1.617.2 lineage) had become the dominant SARS-CoV-2 strain. Parent/caregivers of children younger than 9 years of age were instructed to assist their children in completing the surveys or to record their answers for them if they were unable to do so; patients aged 9 years and older were instructed to complete the questionnaire independently. Participants were not compensated for their participation in the study. Study data were collected and managed online using REDCap (Research Electronic Data Capture) [21,22].

### 2.2. Measures

The initial study survey included questions about patient characteristics (age, sex, type of residence, number of siblings in the home, household income) for the purpose of describing the sample. A question about how school was being conducted (e.g., fully online, hybrid learning, in-person) was asked at each timepoint and converted to a binary variable (“in-person” versus “other”) for some analyses. Participants were asked about current headache frequency (ranging from 1 = “Less than once per month” to 7 = “Daily”), headache intensity (0–100 visual analog scale from “no pain” to “most pain possible”), and frequency of using as-needed medications (ranging from 1 = “Less than once per month” to 7 = “Daily”) at each timepoint. In the initial survey, participants were also asked to indicate headache frequency, headache intensity, and frequency of use of as-needed medications prior to the onset of the pandemic and to indicate if they perceived their headaches were worse in frequency and intensity now relative to pre-pandemic levels. The 6-item Headache Impact Test (HIT-6) was included at each timepoint to quantify current levels of headache-related disability [23]. The internal consistency reliability (Cronbach alpha) for the HIT-6 for the current sample for the baseline timepoint was 0.87.

To evaluate changes in lifestyle behaviors/routines over the course of the pandemic, a 6-item questionnaire was developed. Participants were asked to use a 5-point Likert scale (1 = “Strongly Disagree” to 5 = “Strongly Agree”) to respond to questions about computer use, sleep, school stress, home stress, challenges of being out of usual activities, and worries about family health. Responses were converted to a binary variable (1–3 = disagree, 4–5 = agree) for analyses. The overall level of psychological stress at each timepoint was assessed using the total mean score of the 12-item Stress in Children (SiC) Questionnaire [24]. Sample items from the SiC Questionnaire include “It is easy to concentrate during lessons at school” and “Sometimes I can’t manage with the things I have to do”. The response metric ranges from 1 (“never”) to 4 (“very often”). The internal consistency (Cronbach alpha) for the current sample for the baseline SiC Questionnaire was 0.71.

Additionally, we used the Coping Self-Efficacy Scale to evaluate coping efficacy/resilience throughout the pandemic [25]. The scale asks respondents to indicate on a 0–100 visual analog scale (with anchors “cannot do at all” to “certain I can do”) their confidence in using various active coping strategies (e.g., “break an upsetting problem down into smaller parts,” “take your mind off unpleasant thoughts,” and “get emotional support from friends and family”). The total mean score on this questionnaire from each timepoint was used for analyses. The Cronbach alpha for the scale administered at baseline was 0.88 for the current sample.

### 2.3. Analyses

Descriptive statistics (frequency counts and percentages, mean or median, standard deviation, ranges as applicable) were used to summarize the data on sample characteristics, headache characteristics, and responses to items pertaining to changes in routine and lifestyle. The Friedman nonparametric test (reported as a chi-square test statistic) was used to evaluate differences across survey timepoints in responses to the questions about changes in routines and lifestyle behaviors.

The Wilcoxon matched pairs signed rank test was used to evaluate if current headache frequency, headache intensity, and abortive medication use reported at the initial survey (T0) was statistically different from perceived headache frequency and intensity prior to the onset of the pandemic. Chi-square analyses of initial survey data were used to evaluate associations of perceived differences in headache frequency and intensity since the pandemic started (improved, worsened, unchanged) with demographic variables (age group, sex, household income group), school status, and headache diagnosis. Similarly, chi-square analyses of initial survey data were used to evaluate associations of perceived changes in headache intensity and frequency with responses to the lifestyle/routines questions.

Changes in headache frequency, intensity, and impact (HIT-6 scores) across timepoints were evaluated using multilevel growth models, with time (in weeks since the initial survey) modeled as a randomly varying predictor of linear change in these variables [26]. Multilevel growth models are considered powerful approaches to modeling change over time by accounting for unequal spacing between assessment intervals, using likelihood-based estimation to incorporate all available data into estimating model parameters, and allowing for the evaluation of both static and time-varying covariates [26]. Assumptions of these models, including linearity, homoscedasticity, and normal distribution of residuals, were first determined to be adequately met using visual inspection of plots (i.e., Q–Q plots and plots of model residuals against predictors) and statistical tests (Levene’s test for homoscedasticity, the Shapiro–Wilk test for normality) [26]. A first-order autoregressive covariance structure was specified for the multilevel analyses based on comparative fit tests [26]. Following initial unadjusted models, the following variables were evaluated sequentially as fixed predictors of time-related change in headache frequency, intensity, and HIT-6 scores: baseline demographic and clinical variables (age, sex, income group, headache diagnosis, school status), responses to the lifestyle/routines questions, and total mean scores on the Stress in Children questionnaire and the Coping Efficacy questionnaire. Multilevel model results are presented as model *b*-coefficients (±standard error) and *t*-tests for statistical significance.

## 3. Results

### 3.1. Sample Characteristics

One-hundred and thirty participants agreed to complete the initial survey, of which 85 (65%) were female. The mean participant age was 13.6 years (SD = 3.8, range 4–20 years). The median category for combined annual household income was USD 75–100 K (range <USD 25,000 to >USD 150,000). The number of siblings in the home ranged from 0–7 (median = 1). The grade level for participants ranged from kindergarten to college (median = 9th grade). Approximately half (n = 67, 51.5%) had a migraine diagnosis, of which 51 (39.2%) had episodic migraines and 16 (12.3%) had chronic migraines. Other less common diagnoses included tension-type headaches, occipital neuralgia, and other primary headache disorders (e.g., primary stabbing headache). Baseline scores on the HIT-6 were most commonly in the “severe headache impact” category (n = 81, 62.3%); scores were in the “little to no impact” range for 15.4%, “some impact” range for 11.5%, and “substantial impact” range for 10.8% of the sample.

Of the 130 respondents to the baseline (T0) survey, 59 (45.3%) completed T1, 58 (44.6%) completed T2, and 41 (31.5%) completed T3. Those completing the follow-up surveys did not statistically differ from the baseline sample in proportion of females, average age, proportion having a migraine diagnosis, or in headache characteristics reported at baseline (headache intensity, headache frequency, or HIT-6 score).

### 3.2. Changes in School Structure

Figure 1 shows the frequency of participants in each school format at each survey timepoint. At the time of the initial survey soon after the pandemic onset (T0), participants were most commonly receiving no schooling (n = 64, 49%). By two months later (T1), 10% of the sample still reported no schooling. By T2, all but one participant was back to some form of schooling, most commonly fully online (46%). Finally, most participants (86%) had returned to in-person schooling by the last survey timepoint (T3).

### 3.3. Changes in Lifestyle Behaviors and Routines

Table 1 shows descriptive statistics for perceived changes in lifestyle behaviors and routines at each survey timepoint. The responses spanned the full range regarding perceptions of changes to amount of computer use, school stress, home stress, difficulty being out of activities, health worries, and sleep. Agreement scores for the item “I am using the computer more now than is usual for me” reliably differed by timepoint, increasing initially and then reducing at the long-term follow-up timepoint, χ^2^(2) = 12.10, *p* = 0.01. The responses to the other items about lifestyle behaviors and routines were not statistically different as a function of timepoint.

### 3.4. Comparison of Recalled Pre-Pandemic Headache Characteristics to after Initial Pandemic Onset

Based on data from the initial survey (T0), current headache frequency, headache intensity, and frequency of use of abortive medication early on in the pandemic were not significantly different from respondents’ perceived status on these variables prior to the pandemic, *t*(129) = 0.85, *p* = 0.40; *t*(129) = −1.71, *p* = 0.09, and *t*(112) = −0.14, *p* = 0.89, respectively.

When asked specifically in the initial survey about perceived differences in headache frequency and intensity now versus prior to the pandemic, respondents most commonly perceived no changes (n = 54, 41.5% and n = 75, 57.7%, respectively). About one third (n = 41, 31.5%) and one fifth (n = 29, 22.3%) of respondents perceived a worsening of headache frequency and intensity since prior to the pandemic, whereas about one quarter (n = 35, 26.9%) and one fifth (n = 26, 20.0%) of participants perceived an improvement in headache frequency and intensity, respectively. There was an association of female sex with perceived increase in headache intensity, χ^2^(2) = 13.73, *p* < 0.01; more females (30.6%) than males (6.7%) perceived worsening of headache intensity. Perceived increases in headache frequency and intensity since the pandemic onset were also associated with reporting increased home stress (χ^2^(2) = 7.43, *p* = 0.02; χ^2^(2) = 6.49, *p* = 0.04, respectively). Additionally, there was a trend of perceived increase in headache intensity being associated with more worrisome thoughts about family health (χ^2^(2) = 5.47, *p* = 0.06). Perceived changes in headache frequency and intensity were not associated with other demographic and clinical variables (age group, household income group, headache diagnosis), school status, or whether a respondent reported increased computer use, school stress, stress from being out of activities, or poorer sleep.

### 3.5. Changes in Headache Characteristics over the Course of the Pandemic

Table 2 and Table 3 show descriptive statistics on HIT-6 scores and HIT-6 disability categories by survey timepoint. Table 4 and Table 5 show descriptive statistics on headache frequency and headache intensity by survey timepoint. Average scores for each of these metrics remained notably consistent over time. Disability scores were elevated, on average, at T0 and remained high, with a modest downward trend in the proportion of respondents in the severe impact category by T3. Headache frequency similarly changed very little over time, remaining at around one headache episode per week on average. Headache intensity modestly trended downward over time, from a median of 61/100 (T0) to 54/100 (T3). 

The unadjusted results of multilevel linear growth model analyses similarly indicated stable headache frequency over time, on average (*b* = 0.001 ± 0.002, *t* = −0.087, *p* = 0.93), and a nonsignificant trend of decreasing headache intensity over time (*b* = −0.06 ± 0.03, *t* = −0.1.76, *p* = 0.08). After adjusting for the demographic/clinical variables, lifestyle/routine item responses, and stress and coping questionnaire scores, only stress emerged as a statistically reliable predictor: Higher scores on the Stress in Children questionnaire were associated with higher headache frequency regardless of timepoint, *b* = 0.61 ± 0.27, *t* = 2.288, *p* = 0.02. Scores on the Stress in Children questionnaire also moderated time-related changes in headache intensity: Headache intensity decreased over time when controlling for stress (*b* = −0.45 ± 0.15, *t* = −3.06, *p* < 0.01), but decreased less rapidly for individuals whose stress scores remained relatively high, *b* = 0.18 ± 0.06, *t* = −2.70, *p* = 0.01.

Headache-related impacts on functioning (HIT-6 scores) were estimated through multilevel growth analyses to, on average, decline over time since baseline, but the decline was not statistically reliable, *b* = −0.01 ± 0.12, *t* = −1.15, *p* = 0.26. When adjusting for covariates, only age was found to moderate the time-related changes in headache impact: older children had less rapid decline over time in headache impact, *b* = 0.01 ± 0.00, *t* = −2.12, *p* = 0.03. There was no significant association of changes in HIT-6 scores over time with the other model covariates.

## 4. Discussion

The current study prospectively investigated the extent to which the COVID-19 pandemic and associated lifestyle disruption impacted headache outcomes for youth with primary headache disorders. Symptoms of primary headaches and other chronic/recurring pain conditions generally have been found to reliably worsen with increased stress [27,28], but levels of stress can vary widely in response to the same event [29]. Studying pediatric headache disorders in the context of the COVID-19 pandemic thus enables a means of further informing our understanding of the moderators of the relationship between stressors and headaches in children and adolescents.

We anticipated that changes associated with the COVID-19 pandemic might have improved rather than worsened headache outcomes for some children with primary headaches by virtue of reduced academic burden and pressures to engage in social activities. Indeed, it is known that school-age children tend to have more headaches during the school year compared to summer months [30], and that emergency department visits for headaches increase when students return to school from breaks [31]. For other children and adolescents, however, headaches were expected to be exacerbated related to increased exposure to household stressors and changes to their “buffers” against stress (e.g., reduced social interaction at school and outside of school, reduced involvement in sports and other activities). Further, the general trend toward increasing screen time since the pandemic began, reduced daily structure, and even the use of personal protective equipment (PPE) potentially could worsen headaches for some [32]. Overall, our study findings partly supported both speculations: a minority of youth perceived an improvement in headache status, another subset perceived a worsening, but, most commonly, headaches remained remarkably stable throughout the studied phases of the pandemic.

There have now been a few other studies that have specifically examined how the pandemic impacted children and adolescents affected by headaches or other chronic or recurring pain disorders. In one cross-sectional study of 142 youths with a migraine diagnosis seen in a neurology unit in Northern Italy from March to April 2020, patients who were having worsening headaches prior to the lockdown most commonly were perceived by their parents to be faring better after the lockdown [19]. However, the study results were based exclusively on parent recall and derived from a single timepoint. Given that perceived improvement was highest among those with previously worsening symptoms, the improvements also may have reflected a regression to the mean and might have occurred regardless of the lockdown. In another cross-sectional study also completed during the early phase of the pandemic (from March to April 2020), participants aged 5–18 years old with primary headache were recruited from nine centers in Italy [18]. Nearly half (46%) reported headache improvement overall, with 39% reporting no change, and 15% reporting worsening during the lockdown. Predictors of worsening status included higher age, higher anxiety symptoms, greater duration of having a headache disorder, and greater severity of headaches prior to the lockdown; predictors of improvement included male sex and reduction in school effort. Interestingly, participants’ perceptions about headache changes (or lack thereof) since the lockdown were found to not be associated with geographic region, headache prevention medication, anxiety specifically about COVID-19, and whether the headache disorder was chronic or episodic. In a study conducted in the United States that also evaluated the early phase of the pandemic (April through July 2020), youth with chronic headaches or another chronic pain condition were found, on average, to have stable or improved trajectories of pain, insomnia, depression, and anxiety [20]. Direct exposure to COVID-19 did not moderate these findings, but secondary economic stress predicted worsening symptom trajectories. Taken together, extant research suggests that, at least early in the pandemic, headache status often improved in youth with primary headache disorders.

Consistent with these other studies with pediatric pain populations, headache outcomes and impacts for our sample did not reliably worsen over the pandemic and, at least initially, were perceived to have improved for a sizable subset of respondents. An almost equal proportion of patients in our sample perceived either improvement or worsening in headaches relative to pre-pandemic status, but, most commonly, headaches and their impact were perceived to be relatively unchanged. Because our study included additional waves of data collection, we were also able to monitor trends in headache status beyond the earliest phase of the COVID-19 pandemic. These prospective data suggested relative stability in headache frequency, severity, and headache-related disability throughout the timepoints sampled. Although we observed variability in reported changes in routines and schooling status over the course of the pandemic, the status of these variables, interestingly, was found to not systematically modify headache trajectories. This contrasts with the Italian studies that found a reduction in school effort to be a key predictor of headache improvement [18,19]. Some studies on youth mental health impact of the early lockdown phase of the COVID-19 pandemic also attributed a finding of improved overall well-being to the lockdown [12]. The difference in our study results may be partly due to sampling differing timepoints and to certain changes being relatively transitory. For example, about half of our sample was back to some form of schooling already by the initial survey timepoint, such that any potential buffering effect of reduced academic stress may already have diminished.

Although we did not observe a systematic impact of pandemic-related school changes on headache status, we did identify other correlates. Higher perceived stress overall, higher home stress, greater worries about family health, female sex, and higher age were at least modestly correlated with perceived initial worsening of headaches and/or less positive trajectories in headache outcomes. These putative moderators of pandemic impacts on headache status are similar to those observed in the other studies conducted during the early pandemic period [18,19,20]. The finding that females and older children/adolescents tended to report more of an adverse impact of the pandemic on headache status aligns with other data showing that these subgroups experienced more mental health impacts [8,33,34]. Given the known association between mood and primary headaches [35,36], trends in worsening mental health in these subgroups may have, therefore, also influenced headache outcomes. Our observed association of increased stress with initial perceived worsening of headaches and with headache intensity remaining more elevated over time also adds further support to the role that perceived stress has in the generation and maintenance of primary headaches [37,38,39]. However, unexpectedly, children’s perceived efficacy in coping with stress did not moderate changes in headache status in the current study. Further research to identify which resilience factors promote better headache outcomes when youth are faced with potentially stressful circumstances would be helpful for informing personalized stress-management interventions for this population.

This study has several limitations that should be considered when interpreting results. The survey was sent to patients from a tertiary headache center serving one region of the United States. As such, findings may not generalize to community samples or youth with headache disorders in other regions and countries, especially given the variability in how different areas responded to the pandemic. However, our findings were largely compatible with those from studies conducted in other countries. Further, our sample size, response rate, and demographic diversity were small, further risking problems with generalizability and precluding full exploration of the role of sociodemographic factors. As with most other studies evaluating impacts of the pandemic, we did not have pre-pandemic baseline data and, instead, used recalled perspectives at the time of an initial survey completed early in the pandemic, which may be influenced by recall bias. Additionally, our study only assessed a few of the many possible psychosocial and sociodemographic variables that have been hypothesized to have relevance to primary headache outcomes. Relatedly, although there are now reports indicating that headaches are a common symptom in children and adults infected with COVID-19 [40,41] and may also be a common adverse effect of vaccination [42], we did not investigate the direct impact of COVID-19 infection or vaccination on headache trajectories; thus, we cannot conclude anything about these relationships based on our sample.

## 5. Conclusions

Taken together, and with these limitations in mind, the main conclusion from our study results is that the COVID-19 pandemic and associated life changes for youth with primary headache disorder and their families did not seem to uniformly alter headache outcomes over time. However, female children and youth reporting greater home stress and health worries were most likely to perceive worsening of headaches associated with the start of the pandemic (relative to pre-pandemic status). Additionally, youth with higher scores on a stress measure had more frequent headaches across all timepoints studied and had less rapid improvement in headache intensity over time. These findings emphasize the complex and idiographic relationship of stressful life events and lifestyle factors with primary headache disorders and underscore the need for further research to inform personalized effective multimodal treatment.

## Figures and Tables

**Figure 1 children-10-00184-f001:**
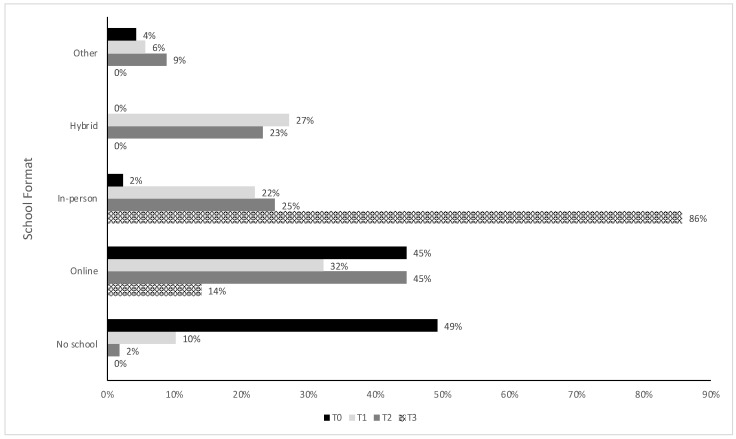
School format for participants by survey timepoint. “Other” comprises those reporting home-schooling and schooling by mailed packets.

**Table 1 children-10-00184-t001:** Descriptive statistics on items about changes in routines and lifestyle by survey timepoint.

	Baseline (T0)			T1			T2			T3		
Lifestyle/Routines Item ^1^	M (SD)	Mdn	Min/ Max	M (SD)	Mdn	Min/ Max	M (SD)	Mdn	Min/ Max	M (SD)	Mdn	Min/ Max
I am using the computer more now than is usual for me	3.45 (1.42)	4.0	1/5	4.02 (1.17)	4.0	1/5	4.13 (1.22)	5.0	1/5	2.67 * (1.02)	3.0	1/4
School is less stressful for me now	3.13 (1.45)	3.0	1/5	2.73 (1.28)	3.0	1/5	2.52 (1.38)	2.5	1/5	2.21 (1.35)	1.5	1/5
Being at home more with family has been stressful	2.68 (1.37)	3.0	1/5	2.77 (1.31)	3.0	1/5	2.94 (1.19)	3.0	1/5	2.53 (1.25)	2.0	1/5
It has been difficult for me to be out of my usual activities	3.68 (1.31)	4.0	1/5	3.80 (1.2)	4.0	1/5	3.79 (1.26)	4.0	1/5	3.25 (1.38)	3.0	1/5
I am having frequent worrisome thoughts about my family’s health and safety	2.98 (1.42)	3.0	1/5	2.97 (1.39)	3.0	1/5	2.89 (1.42)	3.0	1/5	2.69 (1.53)	2.5	1/5
I am getting more sleep now than is usual for me	3.36 (1.36)	4.0	1/5	3.02 (1.14)	3.0	1/5	2.79 (1.21)	3.0	1/5	2.58 (1.53)	2.0	1/5

^1^ Lifestyle/routine items are scored from 1 = Strongly Disagree to 5 = Strongly Agree (3 = Neither agree nor disagree). * Significant (*p* < 0.05) time effect for this item based on the Friedman nonparametric test. Abbreviations: M = mean; SD = standard deviation; Mdn = median; Min/Max = minimum and maximum reported values.

**Table 2 children-10-00184-t002:** Descriptive statistics for HIT-6 total disability score by survey timepoint.

Timepoint	Mean ± SD	Median	Minimum–Maximum
Baseline/T0 (n = 130)	60.21 ± 8.55	62.00	38–78
T1 (n = 59)	59.49 ± 7.70	61.00	40–73
T2 (n = 58)	59.45 ± 7.88	60.00	42–78
T3 (n = 41)	58.44 ± 9.52	59.50	36–74

**Table 3 children-10-00184-t003:** Percentages of participants in HIT-6 headache disability categories by survey timepoint.

Timepoint	Little to No Impact (%, n/N)	Some Impact (%, n/N)	Substantial Impact (%, n/N)	Severe Impact (%, n/N)
Baseline/T0	15.4% (20/130)	11.5% (5/130)	10.8% (14/130)	62.3% (81/130)
T1	10.2% (6/59)	20.3% (12/59)	13.6% (8/59)	55.9% (33/59)
T2	10.3% (6/58)	17.2% (10/58)	15.5% (9/58)	56.9% (33/58)
T3	12.2% (4/41)	14.6% (6/41)	34.1% (14/41)	41.4% (17/41)

**Table 4 children-10-00184-t004:** Descriptive statistics for headache frequency by timepoint.

Timepoint	Mean ^1^ ± Standard Deviation	Median	Minimum– Maximum
Baseline/T0	3.96 ± 1.82	4.00	1–7
Recalled prior to pandemic ^2^	3.85 ± 1.74	3.00	1–7
T1	3.93 ± 1.91	4.00	1–7
T2	4.22 ± 1.75	4.00	1–7
T3	3.97 ± 1.99	4.00	1–7

^1^ Headache frequency is scored as: 1 = Less than once per month; 2= Once per month; 3 = 2–3 times per month; 4 = Once per week; 5= 2–3 times per week; 6= 4–6 times per week; 7= Every day. ^2^ Data for the “recalled prior to pandemic” row are based on respondents’ reporting in the initial survey about perceived headache frequency prior to the onset of the pandemic.

**Table 5 children-10-00184-t005:** Descriptive statistics for headache intensity by timepoint.

Timepoint	Mean ^1^ ± Standard Deviation	Median	Minimum– Maximum
Baseline/T0	56.48 ± 21.00	61.00	0–100
Recalled prior to pandemic ^2^	59.51 ± 20.23	62.00	3–100
T1	56.00 ± 14.52	56.00	22–87
T2	54.34 ± 19.83	58.50	8–91
T3	52.49 ± 22.22	54.00	2–100

^1^ Headache intensity is scored on a 0–100 visual analog scale, with anchors “no pain” to “most pain possible.” ^2^ Data for the “recalled prior to pandemic” row are based on respondents’ reporting in the initial survey about perceived headache intensity prior to the onset of the pandemic.

## Data Availability

A de-identified study dataset can be obtained upon request by contacting the corresponding author.

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
