# Peer review of "A Prospective Evaluation of the Effects of the COVID-19 Pandemic on Youth with Primary Headache Disorders"

_children, 2023, doi:10.3390/children10020184_

Round 1
Reviewer 1 Report
The authors investigated headache-related changes during the Covid-19 pandemic. This study is well-written, easy to follow but it is rather a summary and not an in-depth evaluation without specific hypotheses. 2-year is a long time, and during this period different variants of covid emerged which is not mentioned in the text. This would be important considering that at the beginning children were not at risk.
Abstract:
- Please report statistical results/values (statistical test and p value) in the abstract as well.
Introduction:
- References are needed for the first few sentences as well
- “including tension-type and migraine headache, are chronic health conditions”: I am not sure if tension-type is chronic. Or do the authors talk about chronic tension-type headaches? Please clarify.
- The authors report a study (16) that has found improvement during the lockdown. What type of headache did this study measure?
Also, are there any other studies that investigated headaches, or headache-related symptoms in children during the pandemic? If yes, these studies should be included in the Introduction as well.
Study aim:
- Please add specific hypotheses besides stating the exploration purposes.
Methods:
- Measures: please add the Cronbach alpha values where it is applicable (questionnaires)
- How were the normality and homogeneity tested?
- the youngest participant was 4 years old. I assume the questionnaires had parent-report versions as well. Please clarify this in the Procedure and Measures sections.
- Table 1-6: please add and highlight the significant differences here as well
- “Additionoally” Please correct the typo
Discussion:
- The authors should mention which version of Covid was related to this study period because 1) I believe the later versions might be associated with more headache-related complaints and new daily headaches, 2) children were not at risk at the beginning. Do the authors have information on which participant (or anyone in their household) had Covid during the study period?
Author Response
Response to Reviewer #1 Comments
Thank you for your thoughtful comments about ways to improve our paper. Below, we have restated your comments and have indicated how we have adjusted the paper in response to each:
(1) Please report statistical results/values (statistical test and p value) in the abstract as well.
Response: We had included some statistical results in the Abstract of an earlier draft of the manuscript but were having difficulty maintaining the 200-word limit for the Abstract, so we had ended up removing them on our initially submitted version. Based on your feedback, however, we have now added statistical values back into the Abstract.
(2) References are needed for the first few sentences as well
Response: We have now added reference numbers to make more explicit how each statement made in the opening paragraph of the Introduction has a corresponding reference.
(3) Regarding the statement in the Introduction, “Primary headache disorders in children and adolescents, including tension-type and migraine headache, are chronic health conditions known to be affected by stress, lifestyle factors, and mood” - I am not sure if tension-type is chronic. Please clarify.
Response: In rereading this sentence, we realized that connotations of the term “chronic” can indeed add unintentional confusion. Given that the term “chronic” may be defined differently in different contexts, we elected to just reword this sentence – it now reads, “Primary headache disorders in children and adolescents, including tension-type and migraine headache, are health conditions for which symptom frequency and severity are thought to be moderated by stress, lifestyle factors, and mood.”
(4) The authors report a study (Ref #16) that has found improvement during the lockdown. What type of headache did this study measure? Also, are there any other studies that investigated headaches, or headache-related symptoms in children during the pandemic? If yes, these studies should be included in the Introduction as well.
Response: We have now stated the types of headaches studied (episodic migraine with or without aura, episodic tension-type headache, and chronic migraine) when we mention the Papetti et al study (formerly Reference #16) in the Introduction. As originally described in the Discussion, we noted two additional studies of youth with headaches and pain conditions during the pandemic – one was a cross-sectional study of youth with migraine done in Italy right after the start of the pandemic, and one was a study in the US of youth with chronic pain (with the sample comprised of 64% respondents having ‘chronic headache’ – specific headache diagnoses were not reported in the paper). Based on your comment, in addition to still mentioning these studies in the Discussion section, we have now also mentioned these studies in the Introduction (lines 61-64).
(5) When mentioning study aims, please add specific hypotheses besides stating the exploration purposes.
Response: We have now added a statement of hypotheses to the Introduction section (lines 73-76).
(6) Please add the Cronbach alpha values when mentioning measures, as applicable.
Response: We have now added internal consistency estimates (Cronbach alpha) from our sample data (taken at baseline) when describing our study measures (lines 115-117, lines 127-128, and lines 135-136).
(7) Add information about how normality and homogeneity were tested.
Response: For most of our analyses, we used statistical approaches with less restrictive assumptions about variable distributions (i.e., nonparametric analyses). However, for modeling changes over time in headache variables (frequency, intensity, and headache-related disability), we elected to use a regression-type approach (multilevel growth modeling) to evaluate baseline and time-varying covariates in the models. Assumptions for these multilevel models about linearity, homoscedasticity, and normal distribution of residuals were first evaluated visually (through frequency distributions, plots of model residuals with predictors, and Q-Q plots) as well as quantitatively (Shapiro Wilk and Levene’s tests). We determined through these approaches that the statistical assumptions for multilevel modeling were adequately met for proceeding with these analyses without the need for data transformation. We also compared different options (via a likelihood ratio test) for modeling the covariance structure and ultimately selected an AR1 (first-order autoregressive) structure. We have now added more of these details about what was done prior to the main analyses into the Methods subsection (lines 156-165).
(8) Please clarify in the Procedures if parent-report was used for children in a certain age range.
Response: In the Procedures section (lines 95-98), we have now added further clarification regarding informants. We did not use a parent proxy report per se for the younger children. Rather, for the 12% of our sample who were under the age of 9, parents were instructed to assist their child with reading items and entering answers to the questions as needed (but to still have the child provide their own answers based on their experience and perception).
(9) Please add and highlight significant findings to the study tables.
Response: Where applicable, we have now added an indication of significant findings to the study tables. Note that in most cases of the data reported in the tables, there was no statistically significant result to denote with a symbol, and thus we left the table with the descriptive data reported (while still noting in the text the lack of statistically significant change in the headache variables as a function of time).
(10) Correct the typo in the word “Additionoally.
Response: Thank you for catching this; this has now been corrected.
(11) Please mention which version of COVID was related to the study period, and whether information was available on which participant or household member had COVID during the study period.
Response: Unfortunately, we do not have information about the proportion of patients or family members who had contracted COVID-19 during the study period; see also response to a similar comment from Reviewer #3. From our clinical experience, it was a rare occurrence (<5%) at least during the period of T0-T2 for children seen in our headache program to have been infected with SARS-CoV-2 or to have an infected family member, although it was much more common to have become infected at some point by the final T3 timepoint. Thus, even if we had directly measured infection, we were likely underpowered to evaluate direct effects of infection occurrence on headache trajectories. We have now explicitly mentioned the omission of directly measuring SARS-CoV-2 infection as a limitation. The one other study we are aware of that did look at whether infection with COVIVD-19 altered headache/pain symptoms in children (Law et al., 2021) found infection to not be a significant moderator of symptoms, but that study likely was underpowered as well (as it was done early in the pandemic when child infection rates were still low).
Per your recommendation, we also have now included a statement in the Methods section about the predominant version of COVID during the survey timepoints (section 2.1, lines 87 and 92-95).
Reviewer 2 Report
The paper presented to me for review, "A Prospective Evaluation of the Effects of the COVID-19 Pandemic on Youth with Primary Headache Disorders," prospectively evaluates the effects of the COVID-19 pandemic on headaches in the pediatric population. The paper undoubtedly brings new knowledge to the field of science because data on the long-term effects of the pandemic are still lacking. The authors showed that outcomes of primary headache disorders in youth were not systematically altered by the COVID-19 pandemic.
The paper consists of typical sections, is written in language that the reader can understand, and the results are presented clearly. However, two points should be clarified and highlighted in the manuscript before accepting the paper for publication:
1. we know that primary headaches can be aggravated during the course of infection and after vaccination against SARS-CoV2. Was such information collected in the survey? if so, the results section should be completed, if not, this should be emphasized in the discussion, in which you should write a few sentences on this topic based on the following articles:
https://pubmed.ncbi.nlm.nih.gov/35361131/
https://pubmed.ncbi.nlm.nih.gov/34541916/
2. were respondents assessed over the course of the survey for the development of depressive, anxiety or neurological disorders that may affect headaches, stress levels and quality of life?
Author Response
Responses to Reviewer #2
Thank you for taking the time to review our paper and for the kind comments about the paper’s strengths. Below we have indicated how we have responded to your two specific recommendations for improving the paper.
(1) Please clarify if information about SARS-CoV2 infection or vaccination was collected as part of the study. If so, the results section should be completed, or if not, this should be emphasized in the discussion, in which you should write a few sentences on this topic based on the following articles:
https://pubmed.ncbi.nlm.nih.gov/35361131/
https://pubmed.ncbi.nlm.nih.gov/34541916/
Response: Thank you for this comment; a similar comment was also mentioned by the other reviewers. As our original study focus was primarily on how the changes associated with the pandemic would impact the lives and symptoms of youth with primary headache disorder, unfortunately we did not initially include questions about direct infection or vaccination. At the time the study began, child infection with SARS-CoV-2 was still considered quite rare and vaccines had not yet been developed, so we had not initially thought to study these direct effects as well. We have now explicitly mentioned the omission of directly measuring SARS-CoV-2 infection as a limitation. Additionally, based on your recommendation, we have included the references you suggested regarding potential direct effects of COVID-19 infection and vaccination on headache status (see Discussion section, lines 383-387).
(2) Were respondents assessed over the course of the survey for the development of depressive, anxiety or neurological disorders that may affect headaches, stress levels and quality of life?
Response: Similarly, we did not include formal measures of depression, anxiety, or other neurological/medical conditions that may affect headaches in the study. We do know that all respondents were diagnosed with a primary headache disorder - headaches not attributed to another neurological or medical condition - by headache specialists and had been having headaches for a minimum of several months prior to the start of the study. Our interest in trying to get the study up and running quickly early in the pandemic while minimizing response burden came at the expense of not including a more exhaustive set of biopsychosocial measures. We agree that expanded measures would be helpful for a more complete understanding of pandemic impact in this population, and we have explicitly noted in the limitations section that we only assessed a subset of the many possible psychosocial and sociodemographic variables that might have impacted headache outcomes throughout the pandemic. We anticipate that literature from various regions on impact of the pandemic on primary headaches in children and adolescents will continue to emerge to give an even fuller account of relevant moderating variables.
Reviewer 3 Report
This study included 130 participants (4-20 years) with a grade level ranging from kindergarten to college, and an analysis was focused on the pandemic impact on youth with primary headaches (migraine, tension-type headache, occipital neuralgia, and headache disorders) with an association in the demographics and social features. Headache is the main sign/symptom of COVID-19 in children and adolescents, which impacts this population through lifestyle disruptions. The terms and subclassification of this population are adequately defined in the materials and methods, while the statistical analysis is also described reasonably.
Minor comments
1.- Figure 1: describe the abbreviation FU1, FU2, and FU3. Are these abbreviations T0, T1, T2, and T3??
2.- Changes in school structure: I suggest describing the most important from each time as the author described the baseline time (e.g. T1: 10% of no school and the increase of classes in person from T3)
3.- Check misspellings like “Additinoally” (line 205)
4.- Check the format of p-value (p=.05 or p=0.05)
5.- Can the headache frequency and headache intensity be associated with the SARS-CoV-2 variant? it could be based on the course of the pandemic and the appearance of the VOCs. Several trends have suggested that SARS-CoV-2 variants have caused different symptoms.
6. Tables: All tables have descriptive data, nevertheless these must show the association analysis and I suggest including the p-value in all tables.
7. Conclusion: the conclusion must be focused on the main results:
“There was an association of female sex with a perceived increase in headache intensity”
“Higher scores on the Stress in Children questionnaire were associated with a higher headache”
“Headache intensity decreased over time when controlling for stress”
Author Response
Reviewer #3:
Thank you for the review of our paper and your helpful suggestions about ways to improve the manuscript. Below, we have restated your comments and have indicated how we have integrated your comments into the revised version of our paper:
(1) For Figure 1, please clarify if the abbreviation FU1, FU2, and FU3 was intended to mean T1, T2, and T3.
Response: Thank you for catching this. We have now corrected the labeling of the survey timepoint to be consistent with the rest of the manuscript (T0, T1, T2, and T3).
(2) Consider describing the most important change in school structure at each time point, such as 10% having no schooling at T1 and the increase in in-person classes at T3.
Response: We have now incorporated this suggested change when describing the changes to school structure in Section 3.2 (lines 194-197).
(3) Check misspellings like “Additinoally” (line 205).
Response: This has now been corrected. We also have checked the paper again for other typos that we may have inadvertently missed during our prior editing of the paper.
(4) Check the format of p-values (p=.05 or p=0.05)
Response: We have reviewed our reporting of p-values and adjusted as applicable for consistency. We have reported p-values without a leading 0 before the decimal point (as is typically recommended for reporting in most journals) and have rounded to two decimal places; should a different format be required, we are happy to adjust these again.
(5) Can the headache frequency and headache intensity be associated with the SARS-CoV-2 variant? Several trends have suggested that SARS-CoV-2 variants have caused different symptoms.
Response: We appreciate the speculation about the direct impact of SARS-CoV-2 (and variants) on the headache variables we studied. This point was raised by the other reviewers as well. Although the direct impact of SARS-CoV-2 infection on headache was not an intended focus of our initial study, we agree it would have been interesting to evaluate. In the relative haste of trying to get the study started early on during the pandemic onset, there were measures/items that in retrospect we would have liked to have included but did not – and unfortunately, asking about direct illness with SARS-CoV-2 (by oneself or a family member) was one of those omissions. In our experience, it was a very rare occurrence at least during the period of our T0-T2 surveys for children seen in our headache program to have been infected with SARS-CoV-2, though more common to have been infected at some point by the T3 timepoint. Variant strains also had not yet emerged as predominant during the T0-T2 period of our study, but Delta was the predominant strain in our region by the T3 timepoint (we now have noted this in the paper as well, in response to another reviewer’s comment). Given the low rates of child infection early on, we thus likely would have needed a larger sample size to effectively explore direct impact of SARS-CoV-2 infection on headache status at that time. We have now mentioned the omission of directly measuring SARS-CoV-2 infection as a limitation, given the potential that infection might have had on headache variable trajectories.
(6) Please add and highlight significant findings to the study tables.
Response: Where applicable, we have now added an indication of significant findings to the study tables. Note that in many cases of data reported in the tables, there was not an associated statistical test or statistically significant result to denote with a symbol, such that not every table shows a change. In cases where there was no significant result to report, we still noted this in the text before introducing the table.
(7) The conclusion statement should be focused on the main results, such as ““There was an association of female sex with a perceived increase in headache intensity,” “Higher scores on the Stress in Children questionnaire were associated with a higher headache,” and ““Headache intensity decreased over time when controlling for stress.”
Response: Per your recommendation, we have now expanded the Conclusions statement (section 5.0, lines 392-396) to include reiteration of these other findings.